# Planetary Wave Spectrum in the Stratosphere–Mesosphere during Sudden Stratospheric Warming 2018

Yuke Wang [1], Gennadi Milinevsky [1,2,3,*], Oleksandr Evtushevsky [2], Andrew Klekociuk [4,5], Wei Han [1], Asen Grytsai [2], Oleksandr Antyufeyev [6], Yu Shi [1], Oksana Ivaniha [2,3] and Valerii Shulga [1,6]

1   International Center of Future Science, College of Physics, Jilin University, Changchun 130012, China; wangyk16@mails.jlu.edu.cn (Y.W.); whan@jlu.edu.cn (W.H.); shiyu18@mails.jlu.edu.cn (Y.S.); shulga@rian.kharkov.ua (V.S.)
2   Physics Faculty, Taras Shevchenko National University of Kyiv, 01601 Kyiv, Ukraine; evtush@univ.kiev.ua (O.E.); assen@univ.kiev.ua (A.G.); OksanaIvaniha@univ.net.ua (O.I.)
3   National Antarctic Scientific Center, Department of Atmosphere Physics and Geospace, 01601 Kyiv, Ukraine
4   Antarctic Climate Program, Australian Antarctic Division, Kingston 7050, Australia; Andrew.Klekociuk@awe.gov.au
5   Department of Physics, University of Adelaide, Adelaide 5005, Australia
6   Department of Millimeter Radio Astronomy, Institute of Radio Astronomy, National Academy of Sciences of Ukraine, 61002 Kharkiv, Ukraine; antyuf@rian.kharkov.ua
*   Correspondence: gmilin@univ.kiev.ua; Tel.: +380-50-3525498

**Abstract:** The planetary wave activity in the stratosphere–mesosphere during the Arctic major Sudden Stratospheric Warming (SSW) in February 2018 is discussed on the basis of microwave radiometer (MWR) measurements of carbon monoxide (CO) above Kharkiv, Ukraine (50.0° N, 36.3° E) and the Aura Microwave Limb Sounder (MLS) measurements of CO, temperature and geopotential heights. From the MLS data, eastward and westward migrations of wave 1/wave 2 spectral components were differentiated, to which less attention was paid in previous studies. Abrupt changes in zonal wave spectra occurred with the zonal wind reversal near 10 February 2018. Eastward wave 1 and wave 2 were observed before the SSW onset and disappeared during the SSW event, when westward wave 1 became dominant. Wavelet power spectra of mesospheric CO variations showed statistically significant periods of 20–30 days using both MWR and MLS data. Although westward wave 1 in the mesosphere dominated with the onset of the SSW 2018, it developed independently of stratospheric dynamics. Since the propagation of upward planetary waves was limited in the easterly zonal flow in the stratosphere during SSW, forced planetary waves in the mid-latitude mesosphere may exist due to the instability of the zonal flow.

**Keywords:** planetary wave; mesosphere; stratosphere; major sudden stratospheric warming; microwave radiometer; carbon monoxide; wavelet power spectra

## 1. Introduction

The dynamics of the high-latitude stratosphere in winter is determined mainly by the polar vortex [1,2]. The polar vortex is a stable cyclonic structure with westerly circulation that blocks air mixing in the meridional direction during the polar winter [3]. The vertical structure of the polar vortex determines the penetration of planetary waves (Rossby waves) from the troposphere into the middle atmosphere [4–8].

During high planetary wave activity, the polar vortex is weakened, the zonal circulation may be reversed from westerly to easterly in cases of sudden stratospheric warming (SSW) events [9–14], and the destruction of the vortex is accompanied by a sharp increasing (decreasing) of the polar stratosphere (mesosphere) temperature [1,15–17].

In the Arctic, SSWs are observed regularly, occurring approximately every other winter [17], and in some periods even more often [18]. A typically adopted definition of a

major SSW is the sudden reversal of the zonal wind from westerly to easterly at 10 hPa and 60 degrees latitude [10,11].

The vertical propagation of planetary waves occurs under conditions of moderate westerly wind in the stratosphere [4,5,8]. Conditions of vertical wave propagation are more favorable for waves with small zonal wave numbers *m* [4]. So, the main role in the dynamics of the winter stratosphere and mesosphere is played by zonal waves with *m* = 1 (wave 1) and *m* = 2 (wave 2) [8,13,16,19]. Planetary waves not only propagate upward from the troposphere but can also be generated in the mesosphere due to the instability of the winter polar vortex [19,20] and be sensitive to the evolution of the flow in the low-latitude and equatorial atmosphere [21–23].

The onsets of the recent Arctic SSW events in 2018 and 2019 were associated with vortex split and vortex displacement due to the dominance of wave 2 and wave 1, respectively [12–14]. Along with the altitude and time variation of the amplitude, wave 1 and wave 2 demonstrate changing periodicity (mainly between two and 30 days) at different phases of the SSW event [20,21]. Generally, the main attention in the study of wave 1 and wave 2 was paid to their relative role in the splitting and displacement of polar vortex, as well as to the statistics of periodicity at high latitudes. The altitudinal evolution of the main spectral components of planetary waves in mid-latitudes remains less studied. By making continuous measurements of the mesospheric CO over Kharkiv, Ukraine, with a microwave radiometer (MWR) [12], we are interested in studying the processes associated with regional midlatitude atmospheric dynamics and transport in relation to the SSW. This work also complements knowledge about the vertical and latitudinal dynamical coupling in the stratosphere–mesosphere system during the SSW events [21].

The main goal of this paper was to investigate a planetary wave spectrum in the SSW of 2018. We studied the planetary wave manifestations in the stratosphere and the mesosphere over the Kharkiv region in order to continue analysis of the mesospheric CO from the MWR measurements in Kharkiv [12]. In Section 2, the data and methods of spectral and wavelet analysis of the data are described. In Section 3, the effects of zonal wind reversal during the SSW in zonal anomaly migration are presented, followed by the wave spectrum analysis. The discussion is presented in Section 4 with conclusions in Section 5.

## 2. Materials and Methods

A high-sensitivity microwave radiometer, installed in Kharkiv in 2015 and designed to continuously monitor carbon monoxide (CO) profiles, was used in the winter–spring season 2018 for observation of the SSW effects. For the measurement technique and the results see, e.g., [12,24]. To analyze quantitatively the planetary waves at different pressure levels in the stratosphere and mesosphere, we used the MWR CO data and data on temperature, geopotential heights (GPH) and CO from the Microwave Limb Sounder on the Aura satellite (Aura MLS) [25–28]. Using the Version 4.2 geopotential height data of Aura MLS, the zonal mean zonal wind U was calculated according to the method of Fleming et al. [29]. The grid of GPH data for calculating U was taken with a step of 2° in latitude. If there was a null value, we used a one-dimensional interpolation method to obtain the value of this point. According to [27], the data precision in the troposphere–lower mesosphere (upper mesosphere) is 10–20 ppbv (0.7–11 ppmv) for CO, 0.5–0.8K (1.3–3.6K) for temperature, and 30–45 m (60–110 m) for geopotential height.

For each day, the GPH anomalies were obtained for satellite transitions in motion from the ascending and descending nodes of the orbit at pressure levels between 38 hPa and 0.01 hPa. To obtain these data, a grid with a step of 1° in latitude and 1.5° in longitude was used. Then, a two-dimensional data set of corresponding longitude and time was formed for each pressure level and orbit direction at values averaged in the latitudinal range from 47.5° N to 52.5° N (around the latitude of Kharkiv) [12]. Next, a two-dimensional Fourier analysis was performed for each data set at regular intervals. This analysis allowed us to obtain the periods of waves with zonal wave numbers *m* = 1–5, moving to the east or west.

Using the method described by Torrence and Compo [30], wavelet analysis on the MLS and MWR CO data was performed. The Morlet wavelet as a wavelet basis function was chosen. The Morlet wavelet gives finer resolution in the time–frequency domain than other wavelet functions [30]. A wavelet transform with the mother wavelet $\psi(t) = e^{-t^2/2} \cdot e^{i \cdot 6t}$ was used.

In this work, we compared the gridded Aura MLS CO data with the local MWR CO measurements. As in the case of latitudes (47.5–52.5° N), the selected longitude ranges in the Aura MLS should be as consistent as possible with the location of the ground-based microwave radiometer. The ranges of longitude should be as small as possible and, at same time, the number of days uncovered by the Aura observation should be as few as possible. We had to make a compromise between these two factors. If we took ±2.5° longitude around the desired targeted point, then Aura's observations did not cover this longitude range for nearly half of the time in January to March (~44 days). If the selected longitude ranges increase, then the number of days uncovered by the Aura observation decreases (11 days at ±5.5°). After a compromise selection, the longitude interval ±5.5° was chosen, which gives a cell of 47.5–52.5° N and 30.5–41.5° E in the MLS CO wavelet analysis for the Kharkiv region. Data on the dates without MLS measurements were obtained by one-dimensional interpolation. The optimal choice of the longitude range is important in the wavelet transform, since some short-period fluctuations may be smoothed over a wide range of longitudes.

In this paper, we used the MLS data that covered the main phases of the SSW event in 2018 from 1 January up to 31 March.

## 3. Results

### 3.1. Zonal Wave Migrations

According to the Charney–Drazin criterion [4], large-scale planetary waves propagate from the troposphere into the stratosphere under a moderate westerly zonal flow condition. As seen in Figure 1, the zonal wind underwent strong transformation between the prewarming and warming periods of the SSW 2018. At the Kharkiv latitude 50° N (vertical line), a zonal wind of about 30 ms$^{-1}$ took the height range 40–60 km in January–early February 2018 (Figure 1a,b). This wind speed allowed propagation of the eastward traveling waves as shown below.

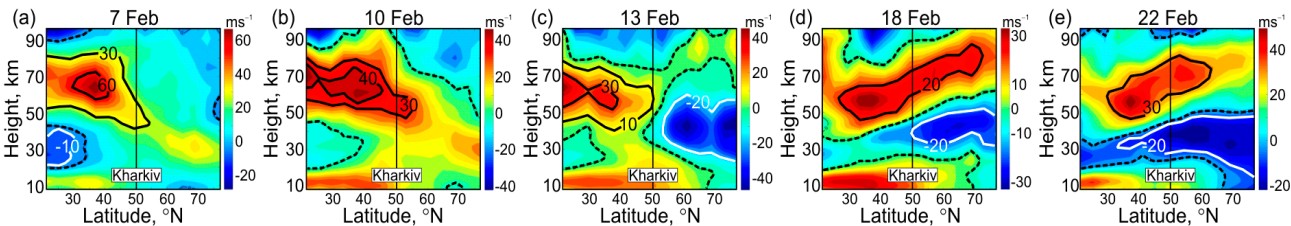

**Figure 1.** Meridional (latitude–height) cross sections of zonal mean zonal wind on (**a–e**) 7, 10, 13, 18 and 22 February 2018 from Aura microwave limb sounder (MLS) data. Dashed line shows zero wind. Solid black and white contours outline westerly and easterly zonal flows, respectively, at several characteristic velocities. Vertical line indicated the Kharkiv latitude 50° N.

A layered wind structure appeared during the SSW: the wind direction reversed to be easterly in the stratosphere, whereas the westerly maximum 20–30 ms$^{-1}$ was displaced upward in the midlatitude–polar mesosphere (60–80 km, Figure 1c–e; see also [12] for zonal wind reversal in the SSW 2018). The easterly zonal flow in the stratosphere became unfavorable for upward wave propagation. Note that he layered zonal wind in the second half of February 2018 extended between the tropical and polar regions (Figure 1d,e). The westerly layer during SSW in Figure 1c–e elevated poleward from the stratopause (~50 km) to the mesopause (80–90 km). This suggests that a possible instability of the zonal flow can develop on a hemispheric scale and forced planetary waves in the polar mesosphere can be

sensitive to the evolution of the flow in the low-latitude and equatorial atmosphere [21–23], as noted in Section 1.

The zonal flow transformation was visible from the time–longitude variations in the MLS geopotential height (Z) anomalies determined with respect to the mean climatology of 2005–2017 for the Kharkiv zone 47.5–52.5° N (Figure 2). The mesospheric and stratospheric levels during January–March 2018 (Figure 2a–e) are presented. Eastward migrating high Z anomalies were observed in the prewarming period (black dashed lines in Figure 2) indicating the traveling planetary wave presence under conditions of westerly zonal wind (Figure 1a,b).

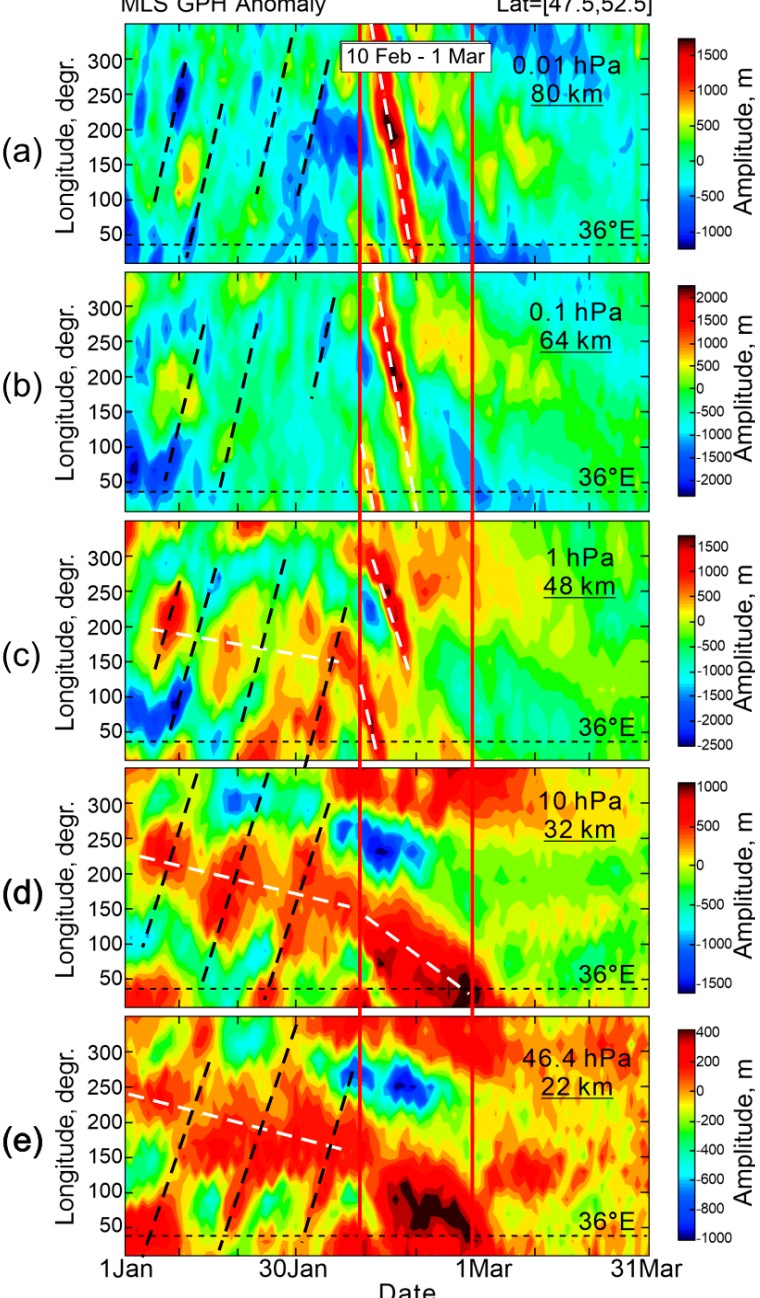

**Figure 2.** (**a**–**e**) Time–longitude variations of the MLS geopotential height anomalies in the Kharkiv zone 47.5–52.5° N during January–March 2018 with respect to the mean climatology between 2005–2017. Red vertical lines mark the sudden stratospheric warming (SSW) event between 10 February and 1 March. Dashed lines indicate eastward (black) and westward (white) high geopotential height anomaly propagation. Dotted horizontal line shows the 36° E longitude of the Kharkiv station.

The sharp change in the direction of the anomaly migration from eastward to westward (black and white dashed lines, respectively) around 10 February is clearly seen (Figure 2). This change coincided with the reversal of the stratospheric westerly to easterly at the Kharkiv latitude at the SSW onset (Figure 1c–e). Corresponding change occurred in conditions for propagation of the traveling planetary waves (Figure 2) in agreement with the Charney–Drazin criterion [4]. Eastward migration in Figure 2 disappeared during the warming and postwarming periods, but westward migration was more distinct in the stratopause–mesosphere region during the SSW event (Figure 2a–c). It is seen also that slowly westward migrating anomalies existed in the stratosphere–stratopause altitudes in the prewarming period (white dashed lines until 10 February in Figure 2c–e). This type of anomaly can be a manifestation of a quasi-stationary planetary wave in the stratosphere.

The westerlies in the prewarming period (Figure 1a,b) provided propagation of the zonal wave 1 and wave 2 with maximum amplitude (derived from the MLS geopotential height) in the stratosphere–lower mesosphere (January–until about 10 February, Figure 3). The stratospheric easterly since mid-February (Figure 1d,e) inhibited upward-propagating planetary waves, as confirmed by change in the wave 1 and wave 2 amplitudes (Figure 3). Wave 2 disappeared during the SSW event and later (Figure 3b). However, a gradually weakening wave 1 was observed at 60–80 km (Figure 3a), at the heights of the westerly maximum (Figure 1d,e, see Section 3.2 for explanation).

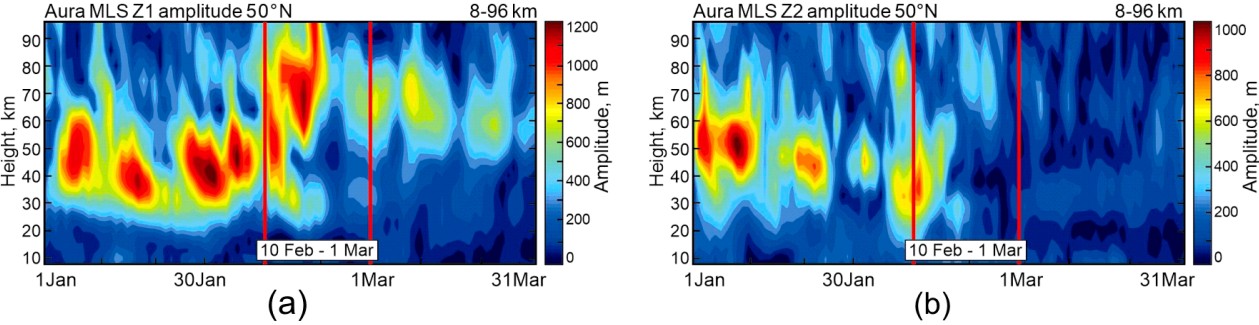

**Figure 3.** Time–altitude section of (**a**) wave 1 and (**b**) wave 2 amplitude in geopotential height at 50° N in January–March 2018 from the MLS data. Vertical lines mark the SSW event between 10 February and 1 March.

Figures 1–3 demonstrate a close relationship between changes in (i) the direction of the zonal wind associated with the SSW, (ii) the migration of zonal anomalies caused by traveling planetary waves and (iii) the amplitudes of zonal wave 1 and wave 2. The effect of the Charney–Drazin criterion [4] on the upward propagation of planetary waves is clearly visible.

### 3.2. Wave Spectrum Changes

As expected from Figures 2 and 3, the wave spectrum should change with the zonal wind reversal. In the lower–middle stratosphere geopotential height change in the anomaly migration direction (Figure 2d,e) was not as pronounced as at the upper levels (Figure 2a–c). The MLS temperature analysis in Figure 4 confirms that zonal anomalies in the stratosphere existed mainly in the prewarming period January to mid-February 2018 (red vertical line at 20 February). In this period, the westerly zonal wind (Figure 1a,b) was favorable for propagation of wave 1 and wave 2 into the stratosphere–lower mesosphere (Figure 3). The slowly westward migrating positive anomaly in the eastern longitudes was a wave-1 ridge of the quasi-stationary planetary wave (white dashed line in Figure 4a–c), which dominated in the lower stratosphere (23 km, Figure 4c).

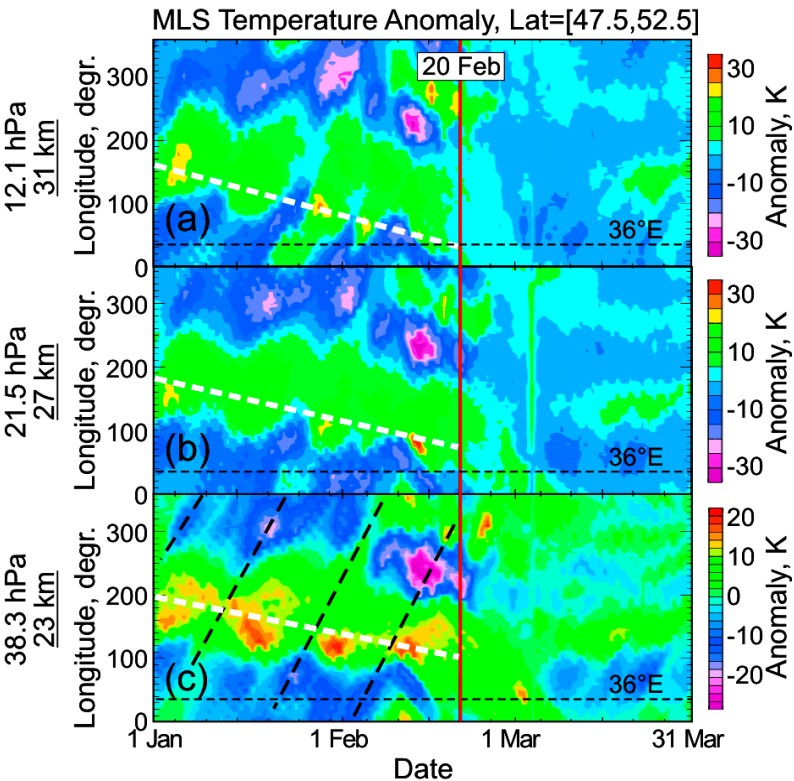

**Figure 4.** As in Figure 2, but for the zonal MLS temperature anomalies in the lower–middle stratosphere at (**a**) 31 km, (**b**) 27 km and (**c**) 23 km; black (white) dashed lines indicate eastward traveling wave 1 (quasi-stationary wave, slowly migrating westward). Red vertical line shows approximate time of the anomaly disappearance in the initial phase of SSW.

Note that the Kharkiv longitude 36° E (dashed horizontal line in Figures 2 and 4a–c) was away from the wave 1 ridge and close to the wave 1 trough during January–March. The wave 1 ridge weakened with altitude and the wave 1 trough became deeper in the western middle stratosphere (31 km, Figure 4a). Negative temperature anomalies were strongest (about –30K) in the initial SSW phase (10–20 February, purple in Figure 4). This is consistent with the splitting of the polar vortex and the displacement of the larger vortex part to the western hemisphere (around 90° W) [12–14].

The vertical wave transformation was accompanied by a westward tilt with altitude seen from sequential westward shift of the wave 1 ridge and trough in Figure 4a–c. By the time of the SSW initial phase (red vertical line in Figure 4), the wave 1 ridge (white dashed line) in the middle stratosphere approached the longitude of Kharkiv (31 km, Figure 4a). At this altitude, the wave 1 ridge was shifted on average by ~50° to the west relative to its longitudinal position in the lower stratosphere (23 km, Figure 4c). The westward phase tilt was consistent with upward propagation of the stationary planetary waves [6] and supports the identification of the sequence of positive temperature anomalies (white line in Figure 4a–c) as the ridge of quasi-stationary wave 1.

It is seen that short periods <5 days are not statistically significant in the stratosphere (Figure 5k–m). The eastward wave 1 (black dashed line in Figure 4a–c) exhibited maximum variance at 10–30 day periods (spectral peaks at $m = 1$ in Figure 5k–m). The westward wave 1 and eastward wave 2 tended to be more intense at the longest periods ($m = -1$ and $m = 2$ in Figure 5k–m), i.e., to be quasi-stationary, as was identified above for $m = -1$.

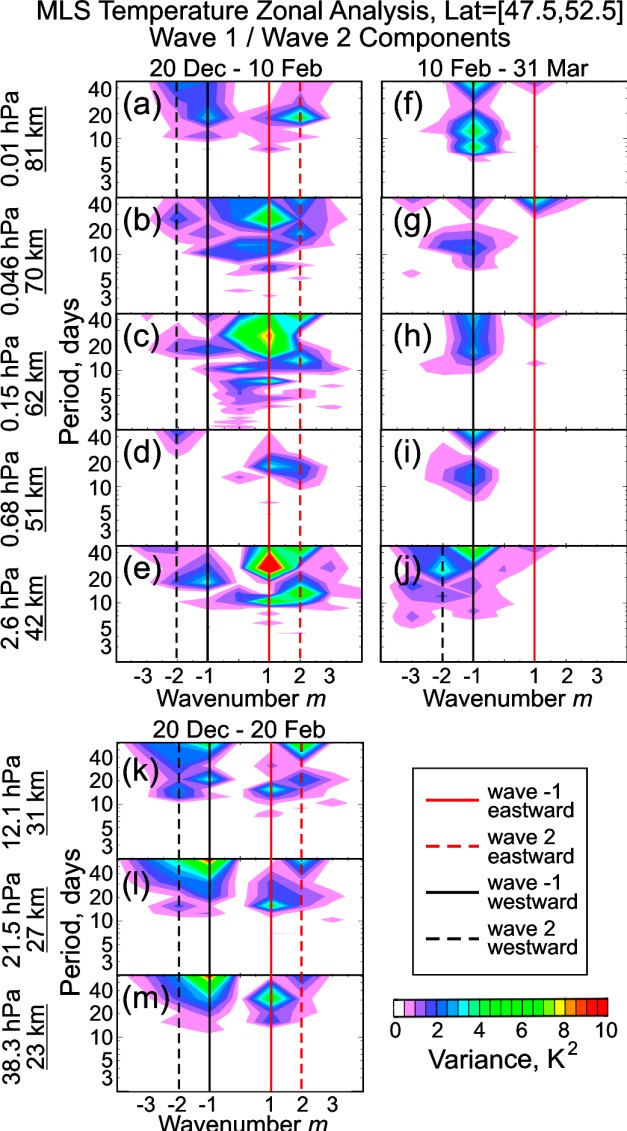

**Figure 5.** The spectral analysis of the zonal temperature anomalies in the upper stratosphere–mesosphere (**a**–**e**) before and (**f**–**j**) after the SSW start on 10 February 2018 and in the lower–middle stratosphere at the pressure levels shown in Figure 4 (**k**–**m**) during 20 December 2017–20 February 2018. Red and black lines indicate the eastward and westward propagating wave numbers $m = 1$ (solid lines) and $m = 2$ (dashed lines). Only variances that are statistically significant at the 95% confidence limit are presented.

To examine the wave spectrum change in the upper stratosphere–mesosphere before and after the SSW 2018 onset (that is suggested from Figures 2 and 3), the two 40-day time intervals are compared in Figure 5a–j. These are 20 December to 10 February and 10 February to 31 March for the pre and postwarming periods, respectively. Figure 5a–j shows five pressure levels between about 40 km and 80 km. The eastward wave 1 demonstrated a maximum spectral signal before the SSW onset ($m = 1$ in Figure 5a–e) and the westward wave 1 dominated after 10 February ($m = -1$ in Figure 5f–j).

This result explains that the prewarming maximum of the wave 1 amplitude at 40–60 km (Z1 in Figure 3a) was formed by the eastward component of $m = 1$, which is also present in the lower stratosphere (Figure 4). At the 60–80 km altitude range, the Z1 maximum in the warming and postwarming periods (Figure 3a) was caused by the westward component of $m = -1$. The transition from eastward to westward propagated wave 1 was noted above from the wave number spectra in Figure 5a–e ($m = 1$ dominates)

and Figure 5f–j (*m* = –1 dominates), respectively. Note that if short and long periods (<5 days and >5 days) are present in the first interval, then the periods longer than 10 days dominate in the second interval (Figure 5a–j, respectively).

Thus, only the longest (*m* = –1) and long-periodic (tens of days) waves dominated in the mid-latitude mesosphere after the zonal wind reversal in the SSW 2018 (Figure 5f–j).

It becomes clear, also, that the prewarming maximum of the wave 2 amplitude in the upper stratosphere–lower mesosphere (Z2 in Figure 3b) appeared due to the eastward component contribution (*m* = 2 in Figure 5a–e), which disappeared later (no spectral signal at *m* = 2 in Figure 5f–j). The wave 2 peak in the lower–middle stratosphere around the SSW onset (red vertical line at 10 February in Figure 3b) was associated with quasistationary wave 2 (*m* = 2 in Figure 5k–m), since traveling wave 2 periods were not extracted here from temperature anomalies.

### 3.3. Wavelet Analysis of Mesospheric CO Variability

In addition to zonal wind, temperature and geopotential height, we examined the wave spectra in the mesospheric CO variability using the MWR and MLS data processed with the wavelet transform (Figure 6). Wavelet analysis revealed changes in the periodic oscillations with time [30].

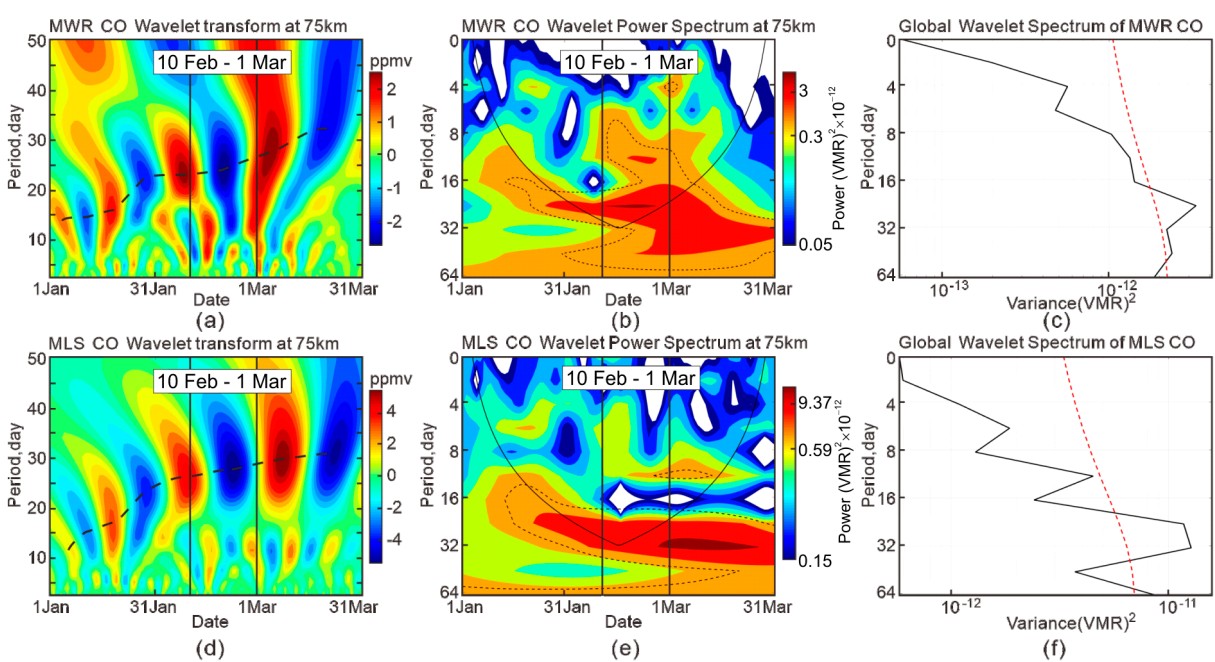

**Figure 6.** The wavelet transform with the real part of the Morlet function for CO from (**a**) microwave radiometer (MWR) data and (**d**) MLS data at 75 km altitude for 1 January – 31 March 2018; (**b**,**e**) power spectrum of the wavelet transform shown in (**a**,**d**) and the solid curve is the cone of influence, within which the spectrum is not significantly influenced by boundary effects. (**c**,**f**) Global wavelet spectrum for (**b**,**e**). Dashed contour in (**b**,**e**) and dashed curve in (**c**,**f**) show the 95% confidence level with the white-noise background spectrum.

We turn our attention to the mesospheric spectral peaks of the eastward wave 1 in the prewarming period (5–30 days at *m* = 1 in Figure 5b,c). They appeared as a sequence of increasing periods in the CO variations during January until the SSW onset from both the MWR and MLS data (Figure 6a,d). Westward wave 1 dominated after the zonal wind reversal (10–30 days at *m* = –1 in Figure 5f–h) exhibiting slowly increasing periods at 20–30-days (early February–March in Figure 6a,d). The wavelet power spectra in Figure 6b,e confirm that a statistically significant signal falling into the cone of influence was observed at 20–30-day periods during the SSW only.

As shown in Figure 6, before the occurrence of SSW, the main periods of CO content at 75 km were ~15–20 days (Figure 6a,b,d,e), and there was also a short period of ~5 days fluc-

tuations. The latter is statistically insignificant in the wavelet power spectrum (Figure 6b,e). After the SSW onset, the smaller period of ~10 days occurred (Figure 6b,e), but the large periods of 20 to 35 days were dominant, which were contributed mainly by westward wave 1 ($m = -1$ in Figure 5f–h). After the end of SSW, the statistically significant periods became larger, although outside the cone of influence (Figure 6d,e), which means that the change of CO approached quasi-stability. Figure 6 shows the periodicities in the daily CO variation at one level of 75 km, and changes in their power were generally consistent with changes in wave 1 and wave 2 at the mesospheric altitudes (Figures 3 and 5).

Wave 2 played an important role in the SSW 2018 and caused the stratospheric polar vortex split [12–14]. As seen at $m = 2$ in Figure 5k–m, wave 2 tended to be quasistationary in the prewarming stratosphere. This is evidence that the quasi-stationary wave 2 was responsible for the polar vortex split in the lower–middle stratosphere in February 2018.

The eastward traveling wave 2 was observed in the prewarming upper stratosphere and mesosphere (15–20-day periods at $m = 2$ in Figure 5a–e) and no traveling wave 2 signal appeared in either the stratosphere and mesosphere after the zonal wind reversal, as noted above in Sections 3.1 and 3.2 (Figure 3b and $m = 2$ in Figure 5f–j).

## 4. Discussion

The impacts of the polar stratosphere dynamics during the SSW 2018 on the mid-latitude stratosphere and mesosphere were analyzed. It was found that abrupt changes in the upward propagation of planetary waves and zonal migration of wave-induced anomalies, as well as in the spectrum and periodicity of waves, occurred near 50° N with the zonal wind reversal after 10 February 2018 (Figures 1–6 in Section 3).

The role of the zonal wave 1 and wave 2 in the SSW preconditioning and development is known from many studies [1,12,31–34]. We showed that wave 1 and wave 2 dominated in the midlatitude stratosphere and lower mesosphere in the prewarming period and at the SSW onset (Figure 3), and that they were quasistationary in their properties ($m = -1$ and $m = 2$ in Figure 5k–m). It was quasistationary wave 2 that was responsible for the splitting of the polar vortex in SSW 2018, consistent with other studies (e.g. [14]), and the manifestation of this large-scale polar process was observed in the midlatitudes. This is not surprising, since from mid-February, zonal wind reversal to the easterlies expanded into the extratropics forming the stratospheric easterly layer between the polar and tropical latitudes (Figure 1c–e).

Zonal wind reversal had a dramatic effect on the planetary wave structure. Upward displacement of the westerly layer into the mesosphere (Figure 1c–e) associated with the elevated stratopause [33] was accompanied by similar changes in the vertical profile of wave 1 (Figure 3a). Wave 1 had two components in the prewarming stratosphere: eastward migrating (black dashed lines in Figure 4c and $m = 1$ in Figure 5k–m) and slowly westward migrating quasistationary (white dashed line in Figure 4a–c and $m = -1$ in Figure 5k–m). In the mesosphere, wave 1 was replaced from eastward migrating to westward migrating with the wind reversal ($m = 1$ and $m = -1$ in Figure 5a–c,f–h, respectively). This change is clearly indicated with dashed lines in Figure 2a,b. Note that such an altitudinal change in zonal migration of wave-induced mesospheric anomalies, as far as we know, has been illustrated for the first time. A separate altitude or altitude interval are usually considered [8,22,35].

Easterly zonal flow in the stratosphere, as noted in Section 3, prevents upward planetary wave propagation according Charney and Drazin [4]. Therefore, the appearance of the westward wave 1 in the mesosphere cannot be caused by tropospheric sources. The simulations made by Limpasuvan et al. [33] showed that westward propagating planetary wave 1 forcing dominates in the polar mesosphere with the SSW onset. The presence of in situ forced planetary waves around the SSW onset due to the eastward (or westerly) jet instability in the polar mesosphere was discussed in [33,35]. Limpasuvan et al. [33] showed that, around the SSW onset, spectral power of the westward wave 1 increases in the polar mesosphere (their Figure 10b). Chandran et al. [35] noted that the reversal of the stratospheric jet at high latitudes in the winter hemisphere leads to conditions conducive

to the generation of short period waves via instability of the background zonal wind field. As distinct from [35], where westward wave 1 in the SSW 2012 had periods <10 days, its dominant periods in the SSW 2018 were 10–20 days (Figure 5f–h). Layered zonal wind structure in Figure 1c–e suggests that zonal flow instability can develop on a hemispheric scale and forced planetary waves in the polar mesosphere can be sensitive to the evolution of the flow in the extratropics [21–23].

Wavelet power spectra showed statistically significant signals in the mesospheric CO variability at the 20–30-day periods (Figure 6). However, if we take the westward wave 1 spectral component, it had maximum spectral power during the SSW 2018 event at the 10–20-day periods (Figure 5f–i). The shorter wave-1 periods of 5–10 days were observed in the prewarming mesosphere (Figure 5a–c). Our results on westward wave 1 suggest that some kind of westerly instability in the midlatitude mesosphere is possible. This possibility needs to be examined in the simulations in the further analysis.

Preliminary estimates showed that the effect of stratopause elevation in the SSW 2018 observed in the wave-1 amplitude (Figure 3a) was absent in the non-SSW event that occurred in 2011 (not shown). The maximum Z1 amplitude in January–March 2011 was stable at the stratopause level and was even higher (1200–1300 m) than in the SSW of 2018 (100–1200 m, Figure 3a), which suggests early final stratospheric warming in spring [36]. Some redistribution of the spectral power of the planetary wave components in the non-SSW winters compared to the SSW winters is possible, which deserves a separate study.

Using MWR data in Kharkiv, mesospheric CO variability was analyzed in the SSW of 2018 and the SSW of 2019 [12,37], in years close to minimum solar activity. Further measurements will be able to show the influence of solar activity on changes in the midlatitude mesosphere [38,39], including changes in the spectral properties of its characteristics.

## 5. Conclusions

The structure and evolution of individual SSWs can differ from event to event [10, 13,18,31,36]. This study presents the changes in the midlatitude planetary wave spectra associated with the SSW of 2018 and their altitudinal dependence. The main conclusion is that the abrupt changes in the wave properties were associated with zonal wind reversal in the stratosphere at the SSW onset. These consistent changes were observed in: (1) the penetration of zonal wave 1 and wave 2 from the troposphere into the stratosphere and mesosphere with likely generation of westward wave 1 in the mesosphere, (2) the direction of the zonal anomaly migration in the mesosphere from eastward to westward, (3) the wave spectrum with the disappearance of wave 1 in the stratosphere and wave 2 in the entire altitude range (10–90 km) and (4) the mesospheric wave 1 periods of 10–20 days with disappearance of the shorter 5–10-day periods observed in the prewarming mesosphere. It should be noted that the mesospheric CO variability in the Kharkiv region in January–March 2018 showed maximum spectral power at 20–30-day periods from both ground-based (MWR data) and satellite (MLS data) observations.

The main advantage of this work is the emphasis on synchronized, but altitude-dependent, changes in the spectral characteristics of planetary waves with the onset of the SSW of 2018.

It is interesting that the two wave 1 components were simultaneously present in the prewarming stratosphere: eastward traveling and quasistationary (slowly migrating westward). Generally, the results are in agreement with other studies and confirm that (i) quasistationary wave 2 was responsible for the stratospheric polar vortex split in February 2018 and (ii) westward wave 1 dominated in the postwarming mesosphere.

It is known from previous work that the westward wave 1 increase in the mesosphere during the SSW may be caused by an unstable westerly polar jet. Based on ground-based and satellite observations, the results of this study showed a change in the characteristic periodicity of mesospheric wave 1 in the SSW of 2018. Although this change was associated with the SSW onset, wave 1 in the mesosphere developed independently of

stratospheric dynamics. The possible flow instability needs to be further explored in simulations considering the midlatitude mesosphere conditions during the SSW.

**Author Contributions:** Conceptualization, A.K., O.E. and G.M.; methodology, O.E., A.K. and Y.W.; data acquisition, A.K., Y.W., V.S., and O.A.; software, A.G., Y.W. and A.K.; validation, A.G., A.K., O.E. and G.M.; investigation, O.E., G.M. and A.K.; writing—original draft preparation, A.G., O.E., Y.W., and G.M.; writing—review and editing, A.G., V.S., O.E., A.K. and G.M.; visualization, O.E., A.G., Y.W. and O.I.; supervision, G.M. and A.K.; project administration, G.M., V.S. and W.H. Each author contributed to the interpretation and discussion of the results and edited the manuscript. All authors have read and agreed to the published version of the manuscript.

**Funding:** This research received no external funding.

**Institutional Review Board Statement:** Not applicable.

**Informed Consent Statement:** Not applicable.

**Data Availability Statement:** The MWR data presented in this study are available on request from the author V.S.

**Acknowledgments:** This work was supported in part by the Institute of Radio Astronomy of the National Academy of Sciences of Ukraine; by Taras Shevchenko National University of Kyiv, projects 19BF051-08 and 20BF051-02; by the College of Physics, International Center of Future Science, Jilin University, China. This work contributed to the National Antarctic Scientific Center of Ukraine research objectives, and contributed to Project 4293 of the Australian Antarctic Program. The microwave radiometer data were processed using the ARTS and Qpack software packages (http://www.radiativetransfer.org/ (accessed on 15 January 2021)). The Aura Microwave Limb Sounder (MLS) measurements of zonal wind, geopotential height, air temperature and CO were obtained from https://mls.jpl.nasa.gov/data/readers.php (accessed on 15 January 2021).

**Conflicts of Interest:** The authors declare no conflict of interest.

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
