# Peer review of "Planetary Wave Spectrum in the Stratosphere–Mesosphere during Sudden Stratospheric Warming 2018"

_remotesensing, doi:10.3390/rs13061190_

Round 1

Reviewer 1 Report

Authors investigated the planetary wave activity in the stratosphere–mesosphere during the Arctic major Sudden Stratospheric Warming (SSW) in February 2018. For their study they used measurements of carbon monoxide (CO) above Kharkiv and the Aura Microwave Limb Sounder (MLS) measurements of CO, temperature and  geopotential heights. 

They found In the mesosphere, that wave 1 was replaced from eastward migrating to westward migrating with the wind reversal.  Authors demonstrated  abrupt changes in the wave properties are associated with zonal wind reversal in the  stratosphere at the SSW onset.

However, authors have to improve the paper and make it more clear and focused on their own findings. For example, in introduction  they describe "In Antarctica, the vortex is more stable, and two sudden stratospheric warming events have so  far been reported in which the wind direction changed to the opposite at 10 hPa and the  stratospheric temperature rose sharply at that altitude. These two events were recorded  in September 2002 [16,19,20] and in September 2019 [21,22]." Whereas the paper is about  planetary wave spectrum in  the Arctic  2018. Thus, it is more important to describe the situation in  the Arctic region and pay attention on situation in 2018.  

Also authors mention polar vortex, but they do not have any results about strength and position of them. Thus, if authors analyse  Planetary Wave Spectrum, they can focus on this topic. For example, it is more important to discuss typical Planetary Wave Spectrum in absence of SSW. 

In Fig. 2 Authors demonstrate  time–longitude variations of the MLS geopotential height anomalies from 1 Jan to 31 Mar. I recommend to use the same time slot in  Fig. 4 a-c for the zonal MLS temperature anomalies. Fig. 4e-4i need to be moved to Fig. 5. Where different time slots analysed differently. 

For example, in Fig. 4d authors demonstrate 3 lines, originated from analysis, shown in Fig. 4g. It will be easier for reader to see the same 3 lines with the same colours in Fig. 4g. The same is also valid for all similar pictures. Why Figures like Fig. 4d are needed? I recommend to remove such plots. Also Figures 5a-5j are hardly to see. 

Authors clearly demonstrate, that the wave structure 10 Feb - 1 Mar is different compare to 1 Jan - 10 Feb and to 1 Mar-31 Mar.  Thus, the period 20 Dec - 10 Feb, used for analysis, corresponds to pre-SSW  phase. however, the period 10 Feb-31 Mar has 2 different parts. Since two-dimensional Fourier analysis can not capture changes in wave pattern, results, shown in Fig. 5p - 5t, are not reliable.    

In Fig. 5i and 5q we see, that the wave amplitudes after 10 Feb are lower than bevor 10 Feb. Figures 6 b, e demonstrate increase of wave activity after 10 Feb, especially for waves with periods between ca. 20 and 32 days. The question is: how can I compare Figures 5i, q with Figures 6b, e?

Also authors have to keep in mind, that observed  behaviour can differ from one SSW to another one. Thus, they need to underline in conclusion, that they report about features SSW 2018. Alternatively, they can  analyse all previous SSWs and conclude how typical such features are for SSWs. 

Author Response

Response to Reviewer 1

Manuscript remotesensing-1110867, MDPI. Planetary Wave Spectrum in the Stratosphere–Mesosphere during Sudden Stratospheric Warming 2018, by Wang et al.

We would like to thank the reviewer for careful and thorough reading our manuscript, as well as for the useful comments and suggestions for improving the text and figures. Changes to the revised manuscript (Rev1) are highlighted in blue. RC – Reviewer’ comments, AC – Authors’ comments.

RC: However, authors have to improve the paper and make it more clear and focused on their own findings. For example, in introduction they describe "In Antarctica, the vortex is more stable, and two sudden stratospheric warming events have so  far been reported in which the wind direction changed to the opposite at 10 hPa and the  stratospheric temperature rose sharply at that altitude. These two events were recorded  in September 2002 [16,19,20] and in September 2019 [21,22]." Whereas the paper is about planetary wave spectrum in  the Arctic  2018. Thus, it is more important to describe the situation in the Arctic region and pay attention on situation in 2018.

AC: Italicized text is removed in Rev1 and sentence “Dissipation of planetary waves in the winter stratosphere warms the polar vortex and reduces its strength [7]. In vertical extent, the waves propagate into the mesosphere, where they break, and also weaken the westerlies [23].” in the next paragraph is also removed as repeating the above.

Lines 42–46 have been corrected to: “During high planetary wave activity, the polar vortex is weakened, and the zonal circulation may be reversed from westerly to easterly in cases of sudden stratospheric warming (SSW) events [9–14] and the destruction of the vortex is accompanied by a sharp increasing (decreasing) of the polar stratosphere (mesosphere) temperature[1,15–17].”

To complement the overview and better highlight the subject of study, a paragraph was added in Introduction, Lines 59–72.

In References, [16, 19–22] replaced by new citations.

RC: Also authors mention polar vortex, but they do not have any results about strength and position of them. Thus, if authors analyse  Planetary Wave Spectrum, they can focus on this topic. For example, it is more important to discuss typical Planetary Wave Spectrum in absence of SSW. 

AC: Polar vortex strength and position in SSW 2018 were analyzed in [12], with our participation. In 2018, wave 2 intensification near 10 Feb (Fig. 3b) results in vortex split with the larger vortex part in the western hemisphere (around 90°W) [12–14]. We note this in Lines 203–206.

More notes to Reviewer 1 and Figures see in attachment

Reviewer 2 Report

This study is trying to derive the wave propagation characteristics during the sudden stratospheric warming in 2018. The results and method used for quite comprehensive and acceptable. However, I do have concerns regarding the need for conducting this study. The authors did not provide any strong reason or motivation why this study was conducted and how this study is going to improve the current understanding of the wave propagation during SSWs. This paper needs to provide a strong motivation along the new findings if any that can add some value. The study should atleast compare with the previous studies and present what this study is contributing to the existing knowledge.  

Here are some specific comments

Line 41 - correct the grammar of the sentence.

Line 60 - the motivation of the study conducted is very weak. What is the need to study the wave spectrum of 2018 event? The wave structures that propagate into middle atmosphere are well known, then what is the missing information that this study is trying to find?

Line 79 - the data quality verification needs to be explained more here. Not just pointing to the reference.

Line 90 – “In the case of the MLS CO data wavelet analysis, the optimal choice of longitude range for analysis is important”. Why is this important?

Line 91-95 - it is hard to understand what the authors are talking about in this sentence. This needs to be re-written.

Line 97 – why was this wavelet chosen?

Author Response

Response to Reviewer 2

Manuscript remotesensing-1110867, MDPI. Planetary Wave Spectrum in the Stratosphere–Mesosphere during Sudden Stratospheric Warming 2018, by Wang et al.

We would like to thank the reviewer for careful and thorough reading our manuscript, as well as for the useful comments, remarks and suggestions. Changes to the revised manuscript (Rev1) are highlighted in blue. RC – Reviewer’ comments, AC – Authors’ comments.

RC: This study is trying to derive the wave propagation characteristics during the sudden stratospheric warming in 2018. The results and method used for quite comprehensive and acceptable. However, I do have concerns regarding the need for conducting this study. The authors did not provide any strong reason or motivation why this study was conducted and how this study is going to improve the current understanding of the wave propagation during SSWs. This paper needs to provide a strong motivation along the new findings if any that can add some value. The study should at least compare with the previous studies and present what this study is contributing to the existing knowledge.

AC: In Lines 63–72 of Rev1, motivation has been clarified based on insufficient knowledge of the vertical evolution of planetary wave spectrum during SSW at mid-latitudes compared to the polar region. By making continuous measurements of the mesospheric CO over Kharkiv with the MWR [12], we are interested in studying the processes associated with regional atmospheric dynamics and transport. This work can help in understanding the vertical and latitudinal dynamical coupling in the stratosphere–mesosphere system during the SSW events (e.g. [21]). The new findings and comparison with the previous studies are highlighted in Lines 70–72, 302–305, 324–327, 339–343, 379–381, 388–392.

RC: Line 41 - correct the grammar of the sentence.

AC: Lines 40–41, corrected to “The vertical structure of the polar vortex determines the penetration of planetary waves (Rossby waves) from the troposphere into the middle atmosphere [4–8].”

RC: Line 60 - the motivation of the study conducted is very weak. What is the need to study the wave spectrum of 2018 event? The wave structures that propagate into middle atmosphere are well known, then what is the missing information that this study is trying to find?

AC: As explained in the first AC above, motivation has been clarified in Lines 63–72.

 RC: Line 79 - the data quality verification needs to be explained more here. Not just pointing to the reference.

AC: We explain data precision in more detail, Lines 92–95: “By [27], the data precision in the troposphere–lower mesosphere (upper mesosphere) is 10–20 ppbv (0.7–11 ppmv) for CO, 0.5–0.8K (1.3–3.6K) for temperature, and 30–45 m (60–110 m) for geopotential height.”

RC: Line 90 – “In the case of the MLS CO data wavelet analysis, the optimal choice of longitude range for analysis is important”. Why is this important?

Line 91-95 - it is hard to understand what the authors are talking about in this sentence. This needs to be re-written.

Line 97 – why was this wavelet chosen?

 AC: Choice of longitude range and wavelet transform is explained in re-written text in Rev1 in new Lines 105–124. In short, longitude range is compromise between the width of the segment, as narrow as possible near Kharkiv, and the coverage of that segment with satellite data.

See also this Response in attached file with the text highlighted in blue. 

Reviewer 3 Report

The paper continues previous studies of authors and considers planetary wave activity in the stratosphere–mesosphere during SSW. The presented results are of scientific interest.

  Comments:

How are your results agreed with the results of work [J.N. Lee, D. L. Wu, A. Ruzmaikin, J. Fontenla, Solar cycle variations in mesospheric carbon monoxide, Journal of Atmospheric and Solar-Terrestrial Physics, 170, 2018, P. 21-34], where the mesospheric variations of CO was connected with solar radiation variability?  The results of J. Lee et al were based on the MLS measurements too.

It will be useful if authors mention correlation of temperature and CO variability with CO2 content in the stratosphere and mesosphere.

The paper can be accepted after revision

Author Response

Response to Reviewer 3

Manuscript remotesensing-1110867, MDPI. Planetary Wave Spectrum in the Stratosphere–Mesosphere during Sudden Stratospheric Warming 2018, by Wang et al.

We thank the reviewer for the useful comments and suggestions. Changes to the revised manuscript (Rev1) are highlighted in blue. RC – Reviewer’ comments, AC – Authors’ comments.

RC: How are your results agreed with the results of work [J.N. Lee, D. L. Wu, A. Ruzmaikin, J. Fontenla, Solar cycle variations in mesospheric carbon monoxide, Journal of Atmospheric and Solar-Terrestrial Physics, 170, 2018, P. 21-34], where the mesospheric variations of CO was connected with solar radiation variability?  The results of J. Lee et al were based on the MLS measurements too.

AC: Cursory comparison shows that daily variations in the mesospheric CO in winter 2018 do not correlate with daily variations in Total Solar Irradiance (TSI; Figure A, lower and middle plots, respectively). Dashed vertical lines indicate positive and negative anomalies in TSI (blue curve in middle plot), which are not associated with CO anomalies (white and yellow curves for regional and zonal mean data). This is evidence that CO variability is caused mainly by dynamical variability induced by planetary waves [12]. The mesospheric CO levels at 50°N, 0.01 hPa (80 km) in 2015 (solar maximum) and 2018 (solar minimum) are close (~10 ppmv), if compare (Lee et al., Feb–Mar in their Fig. 7, lower panel) and [12, Fig. 3e, black curve], that is, without solar cycle modulation.

At the same time, the deep TSI minimum appeared at 10 February, just at the SSW onset. This anomaly may be contributing factor to the stratospheric dynamics with the wave-2 intensification and stratospheric polar vortex split [12]. However, this connection (not yet noted in other studies, as far as we know) is beyond the scope of our work and can be investigated separately.

RC: It will be useful if authors mention correlation of temperature and CO variability with CO2 content in the stratosphere and mesosphere.

AC: Yes, that is useful comment but quick look on this relation does not reveal obvious correlation (Figure B). The TSI, CO and CO2 levels near solar maximum (2013) and solar minimum (2018) show no similarity in daily variations, as well as significant difference in mean values. A much more complete statistical analysis is needed, which should be done in a separate work. We note the importance of such an analysis in Lines 358–362.

More notes and Figures see in attached pdf file.

Round 2

Reviewer 1 Report

Manuscript remotesensing-1110867, MDPI. Planetary Wave Spectrum in the Stratosphere–Mesosphere during Sudden Stratospheric Warming 2018, by Wang et al.

Authors introduced corrections, suggested by first reviewers round. Now manuscript is easier to read.  Conclusions  are supported by the results. I recommend to accept in present form.